# Prevention of Post-Hepatectomy Liver Failure in Cirrhotic Patients Undergoing Minimally Invasive Liver Surgery for HCC: Has the Round Ligament to Be Preserved?

**DOI:** 10.3390/cancers16020364

**Published:** 2024-01-15

**Authors:** Marco Vivarelli, Federico Mocchegiani, Taiga Wakabayashi, Federico Gaudenzi, Daniele Nicolini, Malek A. Al-Omari, Grazia Conte, Alessandra Borgheresi, Andrea Agostini, Roberta Rossi, Yoshiki Fujiyama, Andrea Giovagnoni, Go Wakabayashi, Andrea Benedetti Cacciaguerra

**Affiliations:** 1Hepato-Pancreato-Biliary and Transplant Surgery, Department of Experimental and Clinical Medicine, Polytechnic University of Marche, 60126 Ancona, Italy; vivarelli63@libero.it (M.V.); federico.gaudenzi@ospedaliriuniti.marche.it (F.G.); nicolini_daniele@yahoo.it (D.N.); dott.benedetti@gmail.com (A.B.C.); 2Center for Advanced Treatment of Hepatobiliary and Pancreatic Diseases, Ageo Central General Hospital, Saitama 362-8588, Japan; taiga.wakabayashi@me.com (T.W.); malekamo86@yahoo.com (M.A.A.-O.);; 3Department of Radiology, University Hospital “Azienda Ospedaliera Universitaria delle Marche”, 60126 Ancona, Italy; 4Department of Clinical, Special and Dental Sciences, Università Politecnica delle Marche, 60121 Ancona, Italy

**Keywords:** minimally invasive liver surgery, round ligament, portal hypertension, post-hepatectomy liver failure, ascites, oncologic surgery, HCC

## Abstract

**Simple Summary:**

This international experience investigated the clinical impact of round ligament (RL) preservation during minimally invasive liver surgery (MILS) in cirrhotic patients with mild portal hypertension and borderline liver function. During open surgery, the RL is usually divided in order to facilitate the exposition and mobilization of the liver. On the contrary, during MILS, the surgeons have the chance to preserve the RL when feasible, potentially explaining why there is less post-hepatectomy liver failure reported in Literature in comparison with the traditional open approach. This concept may encourage the use of MILS in patients with hepatocellular carcinoma and cirrhosis, thus expanding indications for radical treatments and for increasing the so-called salvageability (chance of repeated liver resections). Furthermore, this concept may be helpful for patients potentially eligible for future liver transplantation who might benefit from a “surgical downstaging strategy”.

**Abstract:**

Post-hepatectomy liver failure (PHLF) represents a major cause of morbidity and mortality after liver resection. The factors related to PHLF are represented not only by the volume and function of the future liver remnant but also by the severity of portal hypertension. The aim of this study was to assess whether the preservation of the round ligament (RL) may mitigate portal hypertension, thus decreasing the risk of PHLF and ascites in cirrhotic patients while undergoing minimally invasive liver surgery (MILS). All the cirrhotic patients who underwent MILS for HCC from 2016 to 2021 in two international tertiary referral centers were retrospectively analyzed, comparing cases with the RL preserved vs. those with the RL divided. Only patients with cirrhosis ≥ Child A6, portal hypertension, and ICG-R15 > 10% were included. Main postoperative outcomes were compared, and the risk factors for postoperative ascites (severe PHLF, grade B/C) were investigated through a logistic regression. After the application of the selection criteria, a total of 130 MILS patients were identified, with 86 patients with the RL preserved and 44 with the RL divided. The RL-preserved group showed lower incidences of severe PHLF (7.0% vs. 20.5%, *p* = 0.023) and ascites (5.8% vs. 18.2%, *p* = 0.026) in comparison with the RL-divided group. After uni/multivariate analysis, the risk factors related to postoperative ascites were RL division and platelets < 92 × 10^3^/µL, calculated with ROC analysis. The preservation of the round ligament during MILS may mitigate portal hypertension, preventing PHLF and ascites in cirrhotic patients with borderline liver function.

## 1. Introduction

Post-hepatectomy liver failure (PHLF) is a feared complication after hepatic resection, particularly among patients with preoperative impairment of the hepatic function; it represents a major cause of postoperative morbidity and mortality, with an incidence that varies widely in literature [1]. The risk of PHLF is particularly increased when liver resection is carried out in cirrhotic patients with mild portal hypertension, with an incidence of up to 30% [2,3,4].

In the last decade, liver surgery for hepatocellular carcinoma (HCC) in cirrhotic patients has seen a steady increase, as it is demonstrated that resection is an effective treatment, particularly for those patients who are not eligible for liver transplantation [5]. In the last few years, the dramatic expansion of minimally invasive liver surgery (MILS) has increased the proportion of cirrhotic patients with HCC that are offered liver resection, due to the substantial advantages of MILS over the traditional open approach from both the surgical and oncological points of view [5,6,7]. In fact, many studies have demonstrated better short-term outcomes of MILS when compared to open liver surgery, in particular in terms of liver-related complications (i.e., PHLF, bile leak); in the same studies, the rate of radical excision of the tumor did not differ between the minimally invasive and open surgical approaches [8,9,10].

The main factors that contribute to an increased risk of PHLF after liver resection are patient comorbidities, liver impairment, insufficient liver remnant, or inadequate liver function [11]. However, the pathophysiologic mechanism underlying PHLF is still far from being completely disclosed. Acute portal hypertension sometimes develops after hepatic resection and is responsible for what is known as “hyperperfusion syndrome”. Indeed, many preclinical models suggest that portal pressure modulation may improve liver regeneration and, therefore, the risk of developing PHLF [11].

In cirrhotic patients, portal hypertension (pHT) leads to a series of vascular rearrangements aimed at mitigating the hypertension itself, including the creation of veno-venous, porto-systemic shunts. The recanalization of the umbilical vein in the round ligament (RL) is a typical example of this process and represents a radiological sign of significant pHT (Figure 1) [12]. Usually, during open liver surgery, the round ligament is ligated and divided in order to facilitate the mobilization of the liver. On the contrary, during MILS, it is possible to have an adequate view of the operative field, even when the RL is preserved; therefore, some surgeons during MILS are used to preserving the RL entirely down to the umbilicus, while others clip and divide it. However, RL preservation during liver surgery can mitigate the acute portal hypertension that may follow liver resection, reducing the vascular and parenchymal damage of the remnant liver and, therefore, the incidence of PHLF and ascites [13].

The aim of this international multicenter study is to investigate the clinical impact of RL preservation during MILS in cirrhotic patients with mild portal hypertension and borderline liver function. In particular, we assessed whether RL division might be considered a risk factor for postoperative ascites, which represents the most noticeable sign of severe PHLF (ISGLS grade B/C).

## 2. Materials and Methods

A retrospective analysis of a prospectively maintained database of all consecutive liver resections performed in two institutions (Azienda Ospedaliero Universitaria delle Marche—Ancona, Italy; and Ageo Central General Hospital—Saitama, Japan) from 1 January 2016 to 31 December 2021 was performed. Anonymized analysis of data from patients undergoing liver surgery was covered by broad consent approved by the institution’s local ethics committee. All investigations complied with the principles of the Declaration of Helsinki [14].

### 2.1. Study Design

All consecutive cirrhotic patients who underwent MILS due to HCC on cirrhosis were included in the analysis and subsequently divided in two groups: those who had their round ligament divided (“RL-divided group”) at surgery and those whom had their round ligament preserved (“RL-preserved group”). The patients of the two groups who underwent surgery were compared, analyzing baseline characteristics and intra- and postoperative outcomes. Uni- and multivariate logistic regressions were performed in order to assess the risk factors for ascites that we considered an objective sign of severe PHLF, and confounding factors that might bring hypothetical bias were excluded. Finally, a sensitivity analysis excluding patients who had a conversion to open surgery was performed in order to further evaluate its impact on the postoperative course and its relationship with PHLF.

The inclusion criteria considered were adult age (≥18 yo); minimally invasive liver procedures (laparoscopic or robotic); histopathological diagnoses of liver cirrhosis (Ishak stage 6 or Metavir score F4) and of HCC; impaired liver function defined by a Child–Turcotte–Pugh score ≥ A6; and/or an indocyanine green retention rate at 15 min > 10% 12. Moreover, the patients included had evidence of portal hypertension, defined by the presence of at least two of the following clinical parameters: platelet count < 120/nL, splenomegaly >12 cm at CT scan, endoscopic confirmation of esophageal varices, and clinical or radiological evidence of venous collateral shunts (e.g., caput medusae, ascites, and veno-venous shunts at CT scan). On the contrary, open and hand-assisted procedures, major liver resections (≥3 segments), cases with portal vein embolization, planned associating liver partition and portal vein ligation for staged hepatectomy (ALPPS) multivisceral resections, and vascular or biliary reconstruction were excluded from the analysis.

Baseline patients’ characteristics were collected and included age, gender, body mass index (BMI), American Society of Anesthesiologists (ASA) score, indocyanine green retention rate at 15 min (ICG-R15), and previous interventions. Details about the procedures (operative time and estimated blood loss) and the early and late postoperative courses (morbidity, mortality, length of stay, and readmission rate) were also recorded, as well as the incidence and grade of surgical complications. Furthermore, the analysis was conducted on an intention-to-treat basis, with patients requiring conversion to open surgery, maintained in the analysis as a laparoscopic procedure.

### 2.2. Preoperative Assessment

The general preoperative workup included routine blood tests and imaging with triphasic abdominal and thorax computed tomography scans and magnetic resonance imaging with liver-specific contrast when indicated. The LiMON^®^ test was performed when feasible in order to have a direct indicator (PDR and R15) of liver function. The indication for liver resection was evaluated for each patient in a multidisciplinary meeting including surgeons, hepatologists, oncologists, and radiologists.

### 2.3. Surgical Technique and Definitions

Brisbane 2000 nomenclature and its update in 2020 were used to define the localization of the tumoral lesions and the types of resections required [15,16]. Minor liver resection was defined as the resection of less than three Couinaud segments. All the procedures included were performed with curative intent, aiming for R0 resection, except for those patients enrolled for downstaging to liver transplantation [17]. In the present series, 3 patients underwent surgery to downstage the HCC as a bridge to liver transplantation; in these 3 cases, surgery was not aimed at radicality.

The surgical technique was comparable between the two centers. Intraoperative ultrasound was performed in all cases to confirm the number, location, size, and relationship of lesions with major vessels, as well as for guiding parenchymal transection. Anesthetic management involved a restrictive intravenous fluid approach during liver transection, combined with a low central venous pressure. The hepatic pedicle was always surrounded before starting liver resection to be able to carry out the Pringle’s maneuver when needed [18].

The postoperative morbidity was graded according to the Clavien–Dindo classification of surgical complications [19]. The occurrence and severity of PHLF were classified according to the ISGLS definition and grading [1]. Readmission was considered a new hospital admission within 30 days of discharge.

### 2.4. Statistical Analysis

Continuous data were reported as the mean with standard deviation (SD) and compared using a two-sided Student’s t-test for normally distributed parameters. Tests for normality were performed using Kolmogorov–Smirnov and Shapiro–Wilk tests. Continuous, not normally distributed variables were described as median ± interquartile range and compared using the Wilcoxon–Mann–Whitney test. Comparisons between groups for categorical variables were performed using the chi-square test with Yates’ correction or Fisher’s exact test when appropriate.

Univariate logistic regression analysis was performed to investigate risk factors related to postoperative ascites. Subsequently, multivariate logistic regression was conducted for factors with statistical relevance on univariate analysis. The optimal cut-off for continuous variables was obtained from analyses of receiver operating characteristic (ROC) curves [20]. The level of statistical significance was set at *p* < 0.05. All statistical analyses were performed using the Statistical Package for Social Sciences (Version 29.0; SPSS Inc, Chicago, IL, USA).

## 3. Results

Patients operated on from 1 January 2016 to 31 December 2021 in two centers (Ancona, Italy and Ageo, Japan) were reviewed. After application of the inclusion criteria, 130 patients were included in the study. Patients were divided into two groups: 44 underwent MILS with RL division and 86 had the RL preserved (Figure 2).

### 3.1. Preoperative and Intraoperative Characteristics

No significant differences were found in terms of baseline characteristics (age, sex, BMI, ASA score, etiology, grade of cirrhosis, and tumor characteristics) between the two groups; therefore, no statistical matching was needed (Table 1). Interestingly, around ¼ of the patients in both groups had Child–Pugh B cirrhosis. None of the patients presented with portal vein thrombosis at the preoperative work-up. Regarding intraoperative findings, a significantly higher incidence in the conversion rate was observed in the RL-divided group (29.5% vs. 1.6%) compared to the RL-preserved group (*p* < 0.001).

### 3.2. Postoperative Outcomes

Regarding postoperative findings (Table 2), a significantly higher incidence of 30-day postoperative complications was found in the RL-divided group, in comparison to the RL-preserved group [23 (52.3%) vs. 24 (27.9%); *p* = 0.006], while a similar rate of severe complications (grade 3–5) was observed in the two groups. Ninety-day morbidity also differed between the two groups, being higher in the RL-divided cohort [23 (52.3%) vs. 26 (30.2%); *p* = 0.014]. No major septic episodes were observed in our series. Five patients in the whole cohort (3.8%) suffered from intra-abdominal collection, requiring percutaneous drainage placement.

With regard to PHLF, a higher incidence was observed in the RL-divided group, in comparison with the RL-preserved group (29.5% vs. 11.6%, *p* = 0.011); in particular, when considering severe PHLF (grade B–C), the RL-divided group experienced a significantly higher incidence than that of the RL-preserved group (20.5% vs. 7%, *p* = 0.023).

Similarly, patients who underwent liver resection with RL division tended to develop ascites more frequently than those with the RL preserved (18.2% vs. 5.8%, *p* = 0.026). Three patients (2.3%) presented postoperative partial portal vein thrombosis, and only one of these patients, included in the RL-preserved group, experienced post-operative ascites. No significant difference was found in terms of the 30-days readmission rate, ICU stay, and hospital stay between the two groups, and no mortality was reported among the whole population.

### 3.3. Risk Factors for Ascites

Regarding risk factors for postoperative ascites, at the univariate analysis, the ASA 3 score [OR 3.459 (95% CI 1.059 to 11.296); *p* = 0.040]; Child–Pugh B grade of cirrhosis [OR 6.863 (95% CI 2.056 to 22.912); *p* = 0.002]; low platelet count [OR 0.977 (95% CI 0.961 to 0.992); *p* = 0.004]; conversion to open surgery [OR 7.500 (95% CI 2.028 to 27.741); *p* = 0.003]; and the division of the round ligament [OR 3.600 (95% CI 1.101 to 11.766); *p* = 0.034] were predictive of developing postoperative ascites.

After performing multivariate analysis, only low platelet count [OR 0.971 (95% CI 0.954 to 0.989); *p* = 0.002] and the division of the round ligament [OR 6.750 (95% CI 1.730 to 26.336); *p* = 0.006] were confirmed to be independent predictive factors for the development of postoperative ascites (Table 3). The ROC curve analysis found a cut-off value < 92 × 10^3^/µL of the platelets count (AUC = 0.781, *p* ≤ 0.001) to be predictive of postoperative ascites (Appendix A).

### 3.4. Sensitivity Analysis Excluding Patients Converted to Open Surgery

After excluding the patients who underwent a conversion to open surgery, 114 patients (31 belonging to the “RL-divided group” and 85 to the “RL-preserved group”) were considered, and no statistical significance emerged in terms of baseline characteristics (Appendix A). Regarding the post-operative outcome, no difference could be found anymore between the two groups in terms of the 30- and 90-days post-operative complication rates. However, no changes emerged in terms of severe PHLF and ascites, which still differed significantly between the two groups (*p* = 0.049 and *p* = 0.028) (Appendix A). Looking at the risk factors for post-operative ascites, after uni/multivariate analysis, the only variable independently related to it was RL division [OR 4.054 (95% CI 1.016 to 16.181); *p* = 0.047], while the PLT count in this analysis failed to reach statistical significance (Appendix A).

## 4. Discussion

The present study is, to our knowledge, the first one that provides significant clinical evidence to support the hypothesis that the preservation of venous collateral shunts can reduce post-hepatectomy portal hypertension in cirrhotic patients. In the present experience, the preservation of RL improved the postoperative outcome of MILS in cirrhotic patients and reduced the risk of PHLF and ascites, particularly in patients with mild portal hypertension.

Recently, liver surgery in cirrhotic patients has seen a steep development, related mostly to an extensive application of a minimally invasive approach, which has led to less postoperative morbidity and mortality without compromising oncologic results [8,9,10]. However, PHLF still represents a fearful complication after liver surgery in cirrhotic patients, especially in those with borderline liver function. Indeed, its incidence varies in literature, with values ranging from 0.9–5% in patients undergoing hepatectomy with normal liver function and between 5% and 30% in cirrhotic patients [2,3]. Such a variation is mostly related to the heterogenous definition of PHLF reported in literature [21].

To date, one of the most frequently adopted definitions comes from the International Study Group of Liver Surgery (ISGLS). According to it, PHLF is defined as when improvements in the synthesis, excretion, and detoxification functions of the operated liver are not yet observed on post-operative day 5 [1]. This abnormality in blood tests is only one of the first signs that reflect a complex modification in liver homeostasis.

Several possible explanations for the development of PHLF have been proposed in the last decades. Among these, ischemia-reperfusion (IR) damage, small-for-size syndrome (SFSS), and sepsis are the most plausible [22]. In particular, SFSS is a well-recognized complication, not only after certain types of liver transplantation (e.g., living donor or split liver transplantation) but also after liver resection. The core pathophysiologic mechanism seems to be the same in both cases: when the liver mass is significantly reduced, and the portal venous return remains the same, a reciprocal and proportional rise in portal venous pressure might be observed [23]. The acute portal hypertension that develops is therefore responsible for what is known as “hyperperfusion syndrome”; as a consequence, SFSS can be successfully treated by procedures aimed at reducing portal hypertension, including splenic artery ligation, splenectomy, or porto-caval shunt [13]. The “small-for-size” model can be applied to the small remnant liver in particular after major liver resection. Experimental studies demonstrated that a decrease in the portal venous pressure obtained through the diversion of the portal blood flow can mitigate histological damage and prevent PHLF [24,25]. These evidences suggest that portal venous pressure may affect liver regeneration, increasing the risk of developing PHLF [11]. In fact, portal hypertension represents a characteristic sign of cirrhosis and has been associated with increased postoperative morbidity (i.e., PHLF and ascites) and mortality after liver resection (LR), representing, for many years, a formal contraindication for liver surgery [26,27].

Recently, many authors reported that MILS represents an independent predictor of the textbook outcome when dealing with liver surgery for HCC in cirrhotic patients [28,29]. One of the hypothetical reasons underlying this benefit might be related to the capability of MILS in preserving venous and lymphatic collateral shunts typical of cirrhosis with portal hypertension [7]

Taking these elements into account, we investigated the hypothesis that the preservation of the RL during MILS in cirrhotic patients might lead to lower incidences of PHLF and ascites. This concept was just recently investigated by Koliogiannis D. et al. in their case series; however only 10 patients were included, and no comparison was performed with those who had their RL divided [13].

In our series including 130 cases, the RL-preserved group showed a significantly lower incidence of severe (grade B-C) PHLF (7% vs. 20.5%, *p* = 0.023) and postoperative ascites (5.8% vs. 18.2%, *p* = 0.026). These data corroborate the hypothesis that preserving collateral venous shunts in patients suffering from mild portal hypertension may mitigate the hyperperfusion syndrome that seems to be at the basis of PHLF pathogenesis. To rule out possible confounding factors, we investigated the risk factors independently related to postoperative ascites, which was considered the most noticeable sign of severe PHLF (grade B/C). After the univariate analysis, ASA score 3, Child–Pugh B cirrhosis, low platelet count, conversion to the open approach, and the division of RL were predictive of postoperative ascites. However, after the multivariate analysis, only low platelet count and RL division were confirmed to be independent risk factors for developing postoperative ascites. These findings seem to strengthen our initial hypothesis. In addition, throughout ROC curve analysis, we assessed a cut-off value for the low platelet count predictive of postoperative ascites [< 92 × 10^3^/µL (AUC = 0.781, *p* ≤ 0.001)]. This value is comparable to that defined in literature as mild thrombocytopenia (75–150 × 10^3^/µL), which is a characteristic expression in patients with chronic liver disease [30,31]. Therefore, considering low platelet count as an indirect sign of portal hypertension, we can also understand how critical it can be in predicting the risks of PHLF and ascites in cirrhotic patients.

Indeed, during open surgery, the RL is ligated and divided in order to facilitate the exposition and mobilization of the liver. On the contrary, during MILS, the surgeons have the chance to preserve the RL when feasible, potentially explaining why less PHLF is reported in literature, in comparison with the traditional open approach [8,9]. This concept may encourage the use of MILS in patients with borderline liver function who are potentially eligible for a future liver transplantation, with the consequence of increasing the so-called salvageability (chance of repeated liver resections) [17,32].

Some limitations should be acknowledged for a critical evaluation on this study. First, the retrospective and non-randomized design of the study may be responsible for a risk of selection bias. However, this is the first preliminary experience analyzing the impact of RL ligation during liver surgery. Second, no direct value quantifying preoperative portal hypertension, such as hepatic venous-portal gradient (HVPG) measurement, was available. However, the main direct and indirect parameters assessing liver function and portal hypertension (Child–Pugh classification, ICG-R15, PLT value, splenomegaly, and presence of varices at the CT scan) were reported. Finally, the number of patients included was not as high as other studies in the field of liver surgery in cirrhotic patients. However, the current population reflects only selected patients operated on in a recent period (2016–2022), excluding the biases present in other experiences related to the changes in the surgical technique and technologies. Furthermore, this experience represents, to our knowledge, the widest series of patients with portal hypertension who underwent MILS.

## 5. Conclusions

Our study is the first to corroborate the hypothesis that the preservation of the RL may help to mitigate post-hepatectomy PHT, thus preventing PHLF and ascites. Furthermore, it also confirmed the importance of pursuing a minimally invasive approach when performing liver resection in cirrhotic patients, especially in patients with borderline liver function and mild portal hypertension. Hence, based on the present experience, we suggest that the RL should be preserved when feasible during MILS in cirrhotic patients. This finding certainly needs to be confirmed in further studies and randomized controlled trials using the least invasive procedures possible, such as Doppler sonography and/or MR angiography.

## Figures and Tables

**Figure 1 cancers-16-00364-f001:**
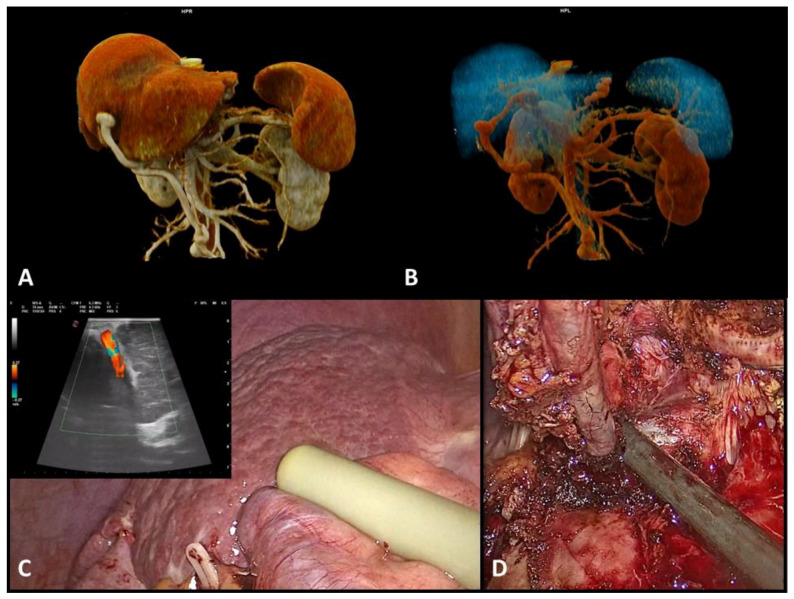
Patients with proven recanalization of the umbilical vein in the round ligament. (**A**,**B**): CT scan 3D rendering; (**C**): US intraoperative of the umbilical vein during MILS in a cirrhotic patient; (**D**): umbilical vein preserved during MILS in a cirrhotic patient. MILS: minimally invasive liver surgery.

**Figure 2 cancers-16-00364-f002:**
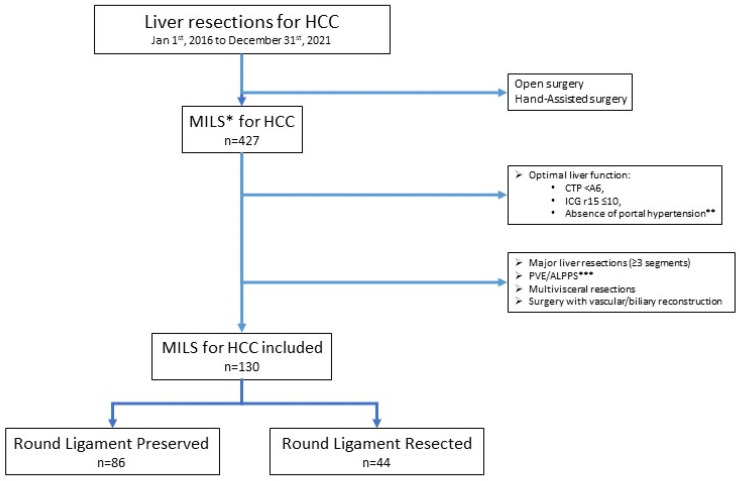
Flowchart of the study population. * MILS: minimally invasive liver surgery; ** PLT > 120/nL, bipolar spleen diameter ≤ 12 cm, or no clinical/radiological evidence of venous collateral shunts; HCC: hepatocellular carcinoma; CTP: Child–Turcotte–Pugh score; ICG: indocyanine green; *** PVE: portal vein embolization; ALPPS: Associating Liver Partition and Portal vein Ligation for Staged hepatectomy.

**Table 1 cancers-16-00364-t001:** Baseline and procedure characteristics of patients undergoing laparoscopic liver resections with round ligament (RL) preservation and RL division.

Patient Characteristics	RL Divided (n = 44)	RL Preserved (n = 86)	*p*
Preoperative Characteristics			
Age, years (IQR)	65 (56–71)	67 (59–76)	0.104
Sex, male (%)	32 (72.7%)	65 (75.6%)	0.723
BMI, kg/m^2^ (IQR)	24 (23–28)	25 (23–29)	0.438
ASA grade			0.281
ASA 1 (%)	0	0	
ASA 2 (%)	26 (59.1%)	59 (68.6%)	
ASA 3 (%)	18 (40.9%)	27 (31.4%)	
ASA 4 (%)	0	0	
Grade of Cirrhosis			0.370
Child–Pugh A (%)	34 (77.3%)	72 (83.7%)	
Child–Pugh B (%)	10 (22.7%)	14 (16.3%)	
Etiology of Cirrhosis			0.255
Alcohol (%)	2 (4.5%)	15 (17.4%)	
HCV (%)	17 (38.6%)	35 (40.7%)	
HBV (%)	7 (15.9%)	11 (12.8%)	
Metabolic (%)	9 (20.5%)	14 (16.3%)	
Other (%)	9 (20.5%)	11 (12.8%)	
ICG-R15 (IQR)	15.0 (12–18)	16.0 (13–19.5)	0.650
Platelets count ×10^3^/µL (IQR)	105 (87–122)	108 (90–120)	0.218
Previous abdominal surgery (%)	26 (59.1%)	38 (44.2%)	0.108
Previous liver surgery (%)	9 (20.5%)	15 (17.4%)	0.675
Number of nodules (IQR)	1 (1–1)	1 (1–1)	0.904
Max. nodule size, mm (IQR)	30 (15–39)	25 (15–35)	0.305
**Intraoperative Characteristics**			
Type of Resection			0.494
Non-anatomic resection (%)	19 (43.2%)	43 (50.0%)	
Segmentectomy (%)	17 (38.6%)	34 (39.5%)	
Left lateral sectionectomy (%)	6 (13.6%)	5 (5.8%)	
Right posterior sectionectomy (%)	2 (4.5%)	4 (4.7%)	
Conversion to Open (%)	13 (29.5%)	1 (1.6%)	<0.001
Pringle maneuver (%)	34 (77.3%)	74 (86.0%)	0.207
Duration of Pringle man, min (IQR)	37 (18–68)	33 (15–41)	0.785
Operative time, min (IQR)	297 (216–380)	264 (195–300)	0.064
Blood loss, cc (IQR)	100 (30–200)	100 (20–190)	0.696

Abbreviations: ASA, American Society of Anesthesiology; BMI, body mass index; IQR, interquartile range, ICG-R15, indocyanine green—retention rate at 15 min; HCV, hepatitis C virus; HBV, hepatitis B virus.

**Table 2 cancers-16-00364-t002:** Postoperative outcomes of patients undergoing laparoscopic liver resections with round ligament (RL) preservation and RL division.

Postoperative Outcomes	RL Divided (n = 44)	RL Preserved (n = 86)	*p*
Postoperative complication—30 days ¥ (%)	23 (52.3%)	24 (27.9%)	0.006
Severe, grade 3–5 (%)	5 (11.4%)	10 (11.6%)	0.964
Postoperative complication—90 days ¥ (%)	23 (52.3%)	26 (30.2%)	0.014
Post Hepatectomy Liver Failure * (%)	13 (29.5%)	10 (11.6%)	0.011
Severe, grade B–C (%)	9 (20.5%)	6 (7.0%)	0.023
Ascites (%)	8 (18.2%)	5 (5.8%)	0.026
30-days readmission (%)	2 (4.5%)	5 (5.8%)	0.762
ICU Stay, days (IQR)	1 (0–1)	1 (0–1)	0.215
Hospital stays, days (IQR)	5 (4–7)	5 (4–5)	0.551

IQR = interquartile range, * ISGLS definition (International Study Group for Liver Surgery), ¥ Clavien–Dindo classification.

**Table 3 cancers-16-00364-t003:** Uni- and multivariate analyses of risk factors associated with the development of postoperative ascites in the whole population.

Ascites Predictive Factors	Univariate Analysis		Multivariate Analysis	
	Odds Ratio (95% CI)	*p*	Odds Ratio (95% CI)	*p*
Preoperative Characteristics				
Age, years	0.980 (0.931 to 1.031)	0.433		
Sex, male	1.149 (0.296 to 4.457)	0.840		
BMI, kg/m^2^	1.008 (0.875 to 1.161)	0.912		
ASA grade				
*ASA 2*				
* ASA 3*	3.459 (1.059 to 11.296)	0.040	2.753 (0.694 to 10.921)	0.150
Grade of Cirrhosis				
Child–Pugh A				
Child–Pugh B	6.863 (2.056 to 22.912)	0.002	1.715 (0.383 to 7.679)	0.481
Etiology of Cirrhosis				
Alcohol				
HCV	0.726 (0.165 to 3.187)	0.671		
HBV	0.275 (0.026 to 2.940)	0.285		
Metabolic	0.444 (0.066 to 3.010)	0.406		
Other	1.000 (0.999 to 1.002)	0.998		
ICG-R15 (IQR)	1.008 (0.989 to 1.027)	0.418		
Platelets count ×10^3^/µL	0.977 (0.961 to 0.992)	0.004	0.971 (0.954 to 0.989)	0.002
Previous abdominal surgery	0.614 (0.190 to 1.989)	0.416		
Previous liver surgery	0.785 (0.162 to 3.798)	0.764		
Number of nodules	0.208 (0.018 to 2.377)	0.207		
Max. nodule size, mm	0.980 (0.940 to 1.022)	0.350		
**Intraoperative Characteristics**				
Round Ligament divided	3.600 (1.101 to 11.766)	0.034	6.750 (1.730 to 26.336)	0.006
Type of Resection				
Non-anatomic resection				
Segmentectomy	0.999 (0.998 to 1.001)	0.998		
Left lateral sectionectomy	0.999 (0.999 to 1.000)	0.999		
Right posterior sectionectomy	0.501 (0.145 to 1.735)	0.275		
Conversion to Open	7.500 (2.028 to 27.741)	0.003	3.611 (0.633 to 20.598)	0.148
Pringle maneuver	0.646 (0.162 to 2.570)	0.535		
Duration of Pringle man, min	1.000 (0.987 to 1.014)	0.993		
Operative time, min	1.001 (0.995 to 1.006)	0.851		
Blood loss, cc	1.000 (0.998 to 1.002)	0.930		

Abbreviations: ASA, American Society of Anesthesiology; BMI, body mass index; IQR, interquartile range, ICG-R15, indocyanine green—retention rate at 15 min; HCV, hepatitis C virus; HBV, hepatitis B virus.

## Data Availability

The data presented in this study are available upon request from the corresponding author.

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
