# Peer review of "Prevention of Post-Hepatectomy Liver Failure in Cirrhotic Patients Undergoing Minimally Invasive Liver Surgery for HCC: Has the Round Ligament to Be Preserved?"

_cancers, 2024, doi:10.3390/cancers16020364_

Round 1

Reviewer 1 Report

Comments and Suggestions for Authors

A very-well written and very interesting paper presenting the advantages of keeping the fragile balance of portal perfusion over the round ligament in cirrhotic patients. Although the collectives are not large, the results are very interesting and the discussion appropriate.

Three points from my side:

1. How many patients suffered from postoperative thrombosis of the portal vein and how many of them had portal vein thrombosis/partial portal vein thrombosis in the preoperative setting?

2.  Were also further causes diagnosed as a (co-)factor for post-hepatectomy liver failure such as septic episode, new-diagnosed (partial) portal vein thrombosis etc.? 

3. What is the opinion of the authors regarding LiMAx as selection tool in the preoperative setting?

4. Over 75% patients (in RL divided-group) and over 85% of the patients (in RL preserved-group) underwent Pringle manoeuvre. Do you think this could be a risk-factor for postoperative impairment of the hepatic function? What is your opinion regarding this aspect.

Comments on the Quality of English Language

Quality of English is good. Minor-editing is required

Reviewer 2 Report

Comments and Suggestions for Authors

Thank you so much for the opportunity to review this paper.

The authors reported an international multicentre retrospective study to investigate the clinical impact of round ligament (RL) preservation during minimally invasive liver surgery in cirrhotic patients with mild portal hypertension and borderline liver function.

After the application of the selection criteria, total of 130 MILS patients were identified, 86 patients with RL preserved and 44 with RL divided. RL preserved group showed lower incidence of severe: post-hepatectomy liver failure (PHLF) (7.0% vs. 20.5%, P=0.023) and ascites (5.8% vs. 18.2%, P=0.026) in comparison with RL divided group.

The paper is interesting, however three is an important bias.

There is a statistically significant difference between the two groups in terms of conversion rate (29.5% vs 1.6%, P < 0.001) for divided and preserved RL group respectively.

Patients with converted surgery from laparoscopy to laparotomy have a higher rate of PHFL, risk of ascites and 90 days complications.

It is necessary to compared the two groups without these subgroup of patients with conversion to open surgery.

Round 2

Reviewer 1 Report

Comments and Suggestions for Authors

All my review points have been answered. No further critical points from my side